# Longitudinally Heterogeneous Tumor Dose Optimizes Proton Broadbeam, Interlaced Minibeam, and FLASH Therapy

**DOI:** 10.3390/cancers14205162

**Published:** 2022-10-21

**Authors:** Matthias Sammer, Aikaterini Rousseti, Stefanie Girst, Judith Reindl, Günther Dollinger

**Affiliations:** Faculty of Aerospace Engineering, Institute of Applied Physics and Measurement Technology (LRT 2), Universität der Bundeswehr München, D-85577 Neubiberg, Germany

**Keywords:** radiation therapy, particle therapy, minibeam therapy, distal edge tracking, FLASH

## Abstract

**Simple Summary:**

The aim of any kind of external radiation therapy is to control a tumor with the highest possible probability of the lowest possible side effects. Here, we study further opportunities of reducing the side effects of proton therapy by applying longitudinally heterogeneous dose distributions in the tumor respecting the delivery of a minimum prescribed dose. In our simulations, the longitudinally heterogeneous dose distributions show a reduced dose in the healthy tissue already in the case of proton broadbeam irradiations, but a much higher (calculated) mean cell survival in the case of proton minibeam irradiation. This demonstrates its potential to substantially reduce side effects at a simultaneously higher tumor control probability, opening new opportunities of easier application when striving for high dose-rate applications of proton beams (>~10 Gy/s), in order to additionally profit from the so-called FLASH effects.

**Abstract:**

The prerequisite of any radiation therapy modality (X-ray, electron, proton, and heavy ion) is meant to meet at least a minimum prescribed dose at any location in the tumor for the best tumor control. In addition, there is also an upper dose limit within the tumor according to the International Commission on Radiation Units (ICRU) recommendations in order to spare healthy tissue as well as possible. However, healthy tissue may profit from the lower side effects when waving this upper dose limit and allowing a larger heterogeneous dose deposition in the tumor, but maintaining the prescribed minimum dose level, particularly in proton minibeam therapy. Methods: Three different longitudinally heterogeneous proton irradiation modes and a standard spread-out Bragg peak (SOBP) irradiation mode are simulated for their depth-dose curves under the constraint of maintaining a minimum prescribed dose anywhere in the tumor region. Symmetric dose distributions of two opposing directions are overlaid in a 25 cm-thick water phantom containing a 5 cm-thick tumor region. Interlaced planar minibeam dose distributions are compared to those of a broadbeam using the same longitudinal dose profiles. Results and Conclusion: All longitudinally heterogeneous proton irradiation modes show a dose reduction in the healthy tissue compared to the common SOBP mode in the case of broad proton beams. The proton minibeam cases show eventually a much larger mean cell survival and thus a further reduced equivalent uniform dose (EUD) in the healthy tissue than any broadbeam case. In fact, the irradiation mode using only one proton energy from each side shows better sparing capabilities in the healthy tissue than the common spread-out Bragg peak irradiation mode with the option of a better dose fall-off at the tumor edges and an easier technical realization, particularly in view of proton minibeam irradiation at ultra-high dose rates larger than ~10 Gy/s (so-called FLASH irradiation modes).

## 1. Introduction

Beside surgery, chemotherapy and immunotherapy, radiotherapy is one of the main treatment options of solid tumors being applied in about 50% of all cases [1,2]. The goal in curative radiotherapy is to stop the proliferation of any malignant stem cell in the tumor. The main channel of inhibition of cell proliferation is meant to be a direct effect from ionizing radiation requiring at least a prescribed minimum dose to any cell within the tumor volume [3,4], although there might be additional effects from cell-to-cell communication, radiation stimulated immune-reactions, and influences on vascular systems or other means of bystander effects [5]. Thus, following the ICRU (International Commission on Radiation Units) recommendation [6,7] for tumor therapy, any position within the tumor volume should not be under-dosed by less than 95% of a prescribed tumor dose.

Eventually, the prescribed tumor dose is limited by the tolerable dose in the healthy tissue. It is the result of the compromise between tumor control and tolerable side effects. In particular, the dose in the healthy tissue must be kept as low as possible under the constraint of a medically indicated prescribed tumor dose to control the tumor, as recommended e.g., in the German radiation protection act [8]. As a consequence, the ICRU also recommends an upper dose limit of 107% of the prescribed tumor dose to be met at any place in the tumor [6,7]. That upper dose limit originates from the general assumption that an over-dosage of the tumor, although possibly even better suited to control the tumor, also results in additional dose delivery to the healthy tissue increasing the risk of side effects [9]. Nonetheless, the upper dose limit may thus be overruled when side effects can be reduced by a more heterogeneous dose deposition within the tumor exceeding the upper ICRU dose criterion while maintaining the minimum dose requirement at any position in the tumor.

In proton or heavy ion therapy, the maximum dose is deposited at the end of the range in the Bragg peak (Figure 1b, red line). It offers the possibility of a longitudinally heterogeneous dose deposition in the tumor maintaining the prescribed minimum dose at any position in the tumor, but also delivering a reduced dose to the surrounding healthy tissue. In this study, the dose distributions of some prominent cases of heterogeneous dose deposition are simulated by applying proton beams from two opposing directions to a simple tumor model. The simulations elaborate the advantageous dose reduction and thus enhanced cell survival within the healthy tissue, specifically tissue that is close to the tumor edges, when ignoring the upper dose limit, but keeping the prescribed dose within the tumor. When combined with additionally lateral dose heterogeneities in proton minibeam therapy, up to a seven-times higher mean cell survival and a corresponding reduction by a factor smaller than 0.4 of the applied effective unified dose is obtained in the healthy tissue compared to all the broadbeam cases. The investigated longitudinally inhomogeneous dose distributions even enhance the potential to reduce the side effects of proton minibeam irradiation modes.

## 2. Materials and Methods

The irradiation scenario is a 25 cm-thick phantom which contains a 5 cm-thick tumor at the center of the phantom as discussed in [10,11]. Therefore, the irradiated healthy tissue is 10 cm thick from each direction. In this study, broadbeams and planar minibeams applied from two opposing directions are considered using the same mean longitudinal dose distributions.

The dose distributions for interlaced proton planar minibeams are calculated similarly to our previous studies using Matlab [12] where planar minibeams have shown to be the best option of increasing mean cell survival in the healthy tissue when applying longitudinally homogeneous mean depth-dose curves [11]. Here, in addition, longitudinally inhomogeneous dose distributions are considered as detailed in the results section.

The lateral dose distributions of an infinitesimal small proton beam at the entrance of a pure water phantom are extracted from the database LAP-CERR [13,14] with its two-Gaussian approximation also accounting for large-angle scattered particles. The distributions are convolved with an initial Gaussian minibeam profile with a standard deviation, σ_0_ = 0.2 mm. No initial secondary particles from a potential minibeam production are included in the dose distributions representing minibeam production through ion focusing. The increasing beam size with tissue depth is assumed to be purely caused by the multiple Coulomb and nuclear scattering in the tissue, hence neglecting initial beam divergences. Additional beam divergences contribute considerably less to the lateral spread even if short focal lens arrangements for minibeam production are considered [15]. The size of the minibeams as produced by the focusing system is the main parameter defining the mean cell survival at the entrance to the patient, i.e., dominating the tissue sparing effects at and close to the skin [16,17]. However, in deeper tissue the beam spreads and the initial beam size is of less importance as long as it is much smaller at the entrance than the beam size resulting from the additional lateral spread in the deeper layers. The main focus of the present simulation lies on the dose distributions and thus the tissue sparing potential in the deeper layers of the healthy tissue up to the tumor edge where dose limits are the most severe. Hence, lateral dose distributions are presented in a depth range of 50–200 mm.

Since the target volume is in the center of the phantom, the opposing dose distributions are symmetrical. For effectively interlacing the two minibeam fields, the fields are shifted by half of the center-to-center (ctc) distances that the dose maxima of one direction lies at the dose minima of the other direction, thus resulting in a total dose distribution that is the sum of the two calculated dose distributions from the two opposing directions. The dose calculations consider 20 minibeams from each direction and the centered unit cell of the regular pattern is used as the representative dose distribution. The numerical accuracy of the centered unit cell in terms of the prescribed tumor dose D_t_ is better than 2.6 × 10^−7^ D_t_. In this study, we allow heterogeneities within the tumor volume as long as the minimum dose D ≥ 0.975 × D_t_ is applied to the tumor at every location. Due to only considering the planar minibeams, just one lateral dimension needs to be calculated. The second lateral dimension is assumed as unlimited, hence no boundary effects, especially for the tumor, are included. The lateral dose distributions D(x,z) are calculated for every millimeter in the z-direction and, in the case of the minibeam irradiations, laterally in the x-direction perpendicular to the planar minibeams in steps of 0.1 mm.

As a comparison to the minibeam irradiations, laterally homogeneous broadbeam scenarios, which also represent the mean depth-dose curves of the minibeam irradiations, are simulated. The longitudinal dose distributions were calculated for every 0.1 mm. The minimum dose in the tumor corresponds to the minimum prescribed dose independent of broad- or minibeam scenarios. For easier reading, the depth-dose curves are all calculated for a prescribed tumor dose of 10 Gy. However, the dose distributions can be linearly scaled to any other prescribed tumor dose. Dose volume histograms are plotted for two sub-volumes in the healthy tissue close to (10 mm–15 mm) and more distant (45 mm–50 mm) from the tumor as well as a volume covering the tumor site.

In order to compare broadbeam with minibeam irradiations, the mean clonogenic cell survival is evaluated using the prescribed tumor dose of D_t_ = 10 Gy. The 2-dimensional dose distributions

D(x,z) are converted into local clonogenic cell survivals S(x,z) using the linear quadratic model (LQ-model) [18]:(1)$$S\left(x,z\right)=\exp\left(-\alpha D\left(x,z\right)-{\beta D}^{2}\left(x,z\right)\right)$$

The α and β values used are taken the same as in the previous studies [10,19], with α = 0.425 Gy^−1^ and β = 0.048 Gy^−2^, i.e., α/β = 8.9 Gy. The model parameters are taken from the PIDE database [20] as the mean values for human cells including tumor and healthy tissue cells waving any RBE effects in the first approximation. The LQ-model is appropriate as long as the percentage scale of cell survival is considered [11]. Thus, it is a good representation of the cell survival in the healthy tissue in the case of broadbeam irradiation modes where cell survivals of 10–20% are obtained for the considered 10 Gy tumor dose.

In the case of the minibeam irradiations, the mean cell survival values, the S(x,z)-values, are averaged in a transversal direction resulting in a mean cell survival S(z) to be compared with the corresponding broadbeam values. The cell survival at the dose peaks will be underestimated by the used LQ-model (Equation (1)). Since cell survival at the dose peaks is low compared to the dose valleys, it results only in a slight underestimation of the mean cell survival of the minibeam irradiation modes. The mean cell survivals are evaluated in terms of an equivalent uniform dose EUD(z) [21] using the same LQ-model for the different irradiation modes for a prescribed tumor dose of 10 Gy:(2)$$\mathrm{EUD}\left(z\right)=\frac{\alpha }{2\beta }\left(\sqrt{1-\frac{4\beta \ln\left(S_{\mathrm{mean}}\left(z\right)\right)}{\alpha^{2}}}-1\right)$$

This is set into relation to the effective dose level of the spread-out Bragg peak scenario EUD_SOBP_(z) by an equivalent unified dose reduction factor DRF(z) where EUD_SOBP_(z) is taken as the physical dose at this depth.
(3)$$\mathrm{DRF}\left(z\right)=\frac{\mathrm{EUD}\left(z\right)}{{\mathrm{EUD}}_{\mathrm{SOBP}}\left(z\right)}$$

## 3. Results

Four different depth-dose distributions are chosen for both the proton minibeam and broadbeam irradiations of the considered tumor model and applied from two opposing directions. Figure 1 shows the dose distributions obtained from each side and the summed distributions for a prescribed tumor dose of D_t_ = 10 Gy:

**SOBP**: A conventional spread-out Bragg peak is shown in Figure 1a. It covers the tumor homogeneously with the dose from each side. The considered proton energies and their relative weighting (see Table 1) is taken from previous studies [10,11]. The summed dose distribution fluctuates less than ±1% relative to D_t_ ($\left|\frac{D_{\mathrm{SOBP}}}{D_{t}}\right|<$ 1.0%). 

**1E**: Figure 1b shows the distal-edge tracking with only one proton energy. The dose is longitudinally heterogeneous. It is the extreme case of irradiation by only using a single Bragg peak of 146 MeV protons from each side. Due to the overlay from the two opposing directions, the dose minimum, D_t,_ is obtained in the middle of the tumor and at the very edges of the tumor, while higher doses are obtained at any other position in the tumor with a dose maximum of about 1.74 ∙ D_t_ at a distance of 3 mm from the tumor edges_._

**HOM:** Figure 1c shows the distal-edge tracking with proton energies that all have a range such that the single Bragg peaks are in the distal half of the tumor. The depth-dose curve when applied for one direction does not homogeneously cover the tumor. However, the single Bragg peaks are weighted such that the irradiation from two opposite directions lead to a homogeneous depth dose with the same low longitudinal fluctuation as in the SOBP case ($\left|\frac{D_{\mathrm{HOM}}}{D_{t}}\right|<$ 1.0%) within the tumor. 

**OPT:** distal-edge tracking with proton energies that also have a range such that the single Bragg peaks are in the distal half of the tumor (Figure 1d). In contrast to the HOM irradiation mode, the longitudinal dose distribution is no longer homogeneous. It is taken for optimized cell survival in the healthy tissue for the two-sided, interlaced proton minibeam irradiation. It allows the maximizing of the center-to-center distance of the planar proton minibeams under the constraint that D > 0.975 D_t_, where D_t_ = 10 Gy, at every location within the tumor volume.

The used energies and their relative weighting factor of the single proton energies relative to that of maximum energy are listed for the four longitudinal irradiation modes in Table 1.

In addition, for each of the four depth-dose distributions, an optimized irradiation case is calculated utilizing the planar minibeams. The minimum dose criterion is taken here that any location in the tumor volume has to be covered with a dose D > 0.975 D_t_ (here: D_t_ = 10 Gy), as already introduced in [11]. An upper dose limit is not considered allowing highly modulated dose distributions within the tumor volume. Therefore, cold spots can be lifted to fit the minimum dose criterion by increasing the mean dose D_mean_ within the tumor by a dose enhancement factor, f_D,_ relative to the prescribed tumor dose, D_t_, thus D_mean_ = D_t_ ∙ f_D_. Heterogeneities of the dose distributions in the tumor volume arise from fluctuations in the lateral dose distribution of the minibeam irradiation, but also from the longitudinal heterogeneous depth-dose distributions in the case of the 1E and OPT irradiation modes that result in f_D_ > 1 already for the broadbeam cases (Table 2). Every minibeam irradiation presented is optimized in terms of the center-to-center distance (ctc) of the minibeams and the dose enhancement factor, f_D_, in order to obtain maximum mean cell survival in the healthy tissue in front of and behind the tumor (Table 2).

The ctc values are only attributable to the proton minibeam irradiation modes. f_D_ describes the enhancement factor of mean doses relative to the broadbeam SOBP case. The mean dose, D_mean_, the mean cell survivals, and the effective unified dose, EUD, for the 10 Gy prescribed tumor dose are evaluated at 10 mm in front and behind the tumor edges. From the EUD values the dose reduction factor DRF is calculated relative to the broadbeam SOBP case. In case of the proton minibeams the effective beam sizes (d_eff,10%_ and d_eff,50%_) for the 10% and 50% survival at the rim of the minibeams are given. The value Δ gives the distance of dose fall-off from the tumor edge into the healthy tissue where the distal dose fall-off has reached 13% D_t_.

The resulting optimized 2D-dose maps for the proton minibeam irradiations of the four longitudinal irradiation modes are presented in Figure 2. The cross sections of the dose distributions at the tumor edge at z = 100 mm and at the tumor center, z = 125 mm, are plotted in Figure 3. Due to symmetry, the dose distribution at the distal tumor edge, z = 150 mm, is the same as at z = 100 mm but shifted laterally by ctc/2. The OPT minibeam irradiation mode shows the result of combined optimization of the longitudinal and the lateral dose distribution where the dose minimum in the tumor is close to the prescribed dose D_t_ around the tumor edge as well as in the center. The minimum dose levels for the other minibeam irradiation modes are larger than D_t_ towards the tumor center (SOBP, HOM) or near the tumor edge (1E) although it is also at D_t_ directly at the tumor edge for the (1E) mode. Thus, the ctc distance is largest and minibeam sizes are the smallest for the OPT irradiation mode as summarized in Table 2.

The resulting dose volume histograms for the two considered regions in the healthy tissue (Figure 4a) as well as for the tumor region are shown in Figure 4b–e. They demonstrate that any location in the considered regions of the healthy tissue obtains nearly the same dose for all broadbeam irradiation cases while largely spread dose values show up for the minibeam cases with some distinct differences between the different cases. For the considered tumor volume, the histograms show that any location in the tumor receives at least the minimum prescribed doses for all cases, but some higher dose values are obtained for the 1E and OPT broadbeam cases. The minibeam cases show a much larger spread of dose values in the tumor resulting in the higher mean dose values as represented by the f_D_ factor given in Table 2. The tissue sparing capabilities of the different irradiation modes are evaluated at 10 mm apart from the tumor edges in the healthy tissue (Table 2): Any of HOM, 1E, and OPT broadbeam irradiation modes show a dose reduction compared to the SOBP mode. The dose reduction factor is DRF = 0.80 for both the OPT and HOM broadbeam irradiation modes with a slightly enhanced mean tumor dose for OPT with f_D_ = 1.04 in the tumor. A DRF = 0.87 is eventually obtained for the 1E broadbeam mode where the mean tumor dose is already enhanced by f_D_ = 1.2.

In order to evaluate the sparing potential of the different minibeam irradiation modes, the mean cell survival is taken as a biological first-order parameter as already introduced in [11]. The mean cell survivals are plotted in Figure 5 for all considered broadbeam and minibeam cases for D_t_ = 10 Gy. The mean cell survival at 10 mm in front of the tumor is presented for all cases in Table 2. It is evaluated in terms of the effective unified dose EUD as given by Equation (2) and the resulting dose reduction factor DRF according to Equation (3) (see Table 2).

Tissue sparing in the healthy tissue depends, in addition to the mean cell survival, on the effective minibeam sizes although information on the relationship of tissue sparing capabilities on minibeam sizes are scarce [16,17,22,23]. Effective minibeam sizes are largest close to the tumor edges and thus were also studied 10 mm before the tumor edges. The effective diameters are much larger than the full-width half-maximum dose levels since the minibeams only spare the healthy tissue in areas of low doses resulting in high cell survival between the minibeams [11]. Effective diameters given by thresholds of 10% cell survival (d_10%_) and 50% cell survival (d_50%_) are also presented in Table 2 since they are considered to be relevant when discussing beam size effects.

An additional feature is the longitudinal dose fall off from the tumor edges into the healthy tissue. Due to the overshoots of the dose close to the tumor edges, the dose gradients at the tumor edges are high in the case of the OPT and the 1E modes, for both the broadbeam and minibeam. The distance between the tumor edge with D = D_t_ and the point where the dose of the irradiation from the opposing side has dropped to 13% of D_t_ is given as the value Δ in Table 2.

## 4. Discussion

### 4.1. Longitudinally Heterogeneous Broadbeam Irradiation Modes

So-called “Distal edge tracking” [24,25] had been introduced in several ways for proton therapy using a broadbeam with longitudinally inhomogeneous dose distributions from several irradiation directions. In most cases, however, a homogeneous dose deposition was aimed for maintaining the lower, but also the upper, ICRU limit as in the HOM case presented here. The reduced dose in the healthy tissue results mainly from the, on average, higher proton energies used to keep the Bragg peaks as far as possible at the distal edges of the tumor. Homogeneity of the dose in the tumor is achieved by applying the beams from several directions.

The four broadbeam irradiation modes applied from two opposing directions considered here contain irradiation modes (1E and OPT) that result in a summed, longitudinally heterogeneous dose distribution. The dose in the healthy tissue is similarly reduced for the homogeneous HOM and the heterogeneous OPT modes compared to the common SOBP case by DRF_OPT_ = DRF_HOM_ = 0.80 (Figure 1, Table 2) resulting in an enhanced cell survival as calculated for a minimum prescribed tumor dose D_t_ = 10 Gy (Figure 5). Eventually, a slight enhancement in tumor control may result from the enhanced mean dose (f_D_ = 1.04) in the case of the OPT irradiation mode. Despite the large dose overshoots and the 20% enhanced mean tumor dose (f_D_ = 1.20), the 1E mode also results in a reduced dose deposition and a higher cell survival in the healthy tissue (DRF_1E_ = 0.87).

Both longitudinally inhomogeneous irradiation modes, OPT and 1E, show much sharper dose fall-off from the tumor edges into the healthy tissue compared to the HOM and SOBP modes. A good comparison of this dose fall-off is the distance Δ between the tumor edge and the point in the healthy tissue where the dose of the distal side of the one-sided irradiation is reduced to 10% ∙ D_t_. In the case of longitudinally homogeneous irradiation modes, this distance is Δ_SOBP_ = 4.6 mm and Δ_HOM_ = 5.0 mm calculated with an uncertainty of about 0.1 mm from the used longitudinal binning. However, the much-reduced values of Δ_OPT_ = 3.2 mm and Δ_1E_ = 2.1 mm are obtained in the case of the OPT and 1E irradiation modes, respectively. These shorter values result from the steep dose gradients at the tumor edges, the steepest for the 1E option, while the dose gradients at the tumor edges are close to zero in the SOBP and HOM cases. The shorter dose fall-offs make it possible to better spare critical organs when they are lying adjacent to the tumor.

An advantage of the 1E compared to all other modes is that only one energy is needed for the irradiation from one side; thereby, reducing irradiation times with better healthy tissue sparing than the conventional SOBP irradiation modes. In addition, minibeam and FLASH irradiation modes become easier to be realized (see next section).

The over-dosage close to the tumor edge and the steep dose gradients of the longitudinally inhomogeneous irradiation modes OPT and mostly for 1E, may have pros and cons in a real tumor irradiation scenario. A major problem may arise in accurately setting the range of the protons. The tumor treatment may suffer from cold spots at the tumor edge, if the proton ranges are smaller than calculated or, vice-versa, the hot spots are shifted towards the healthy tissues in case of larger proton ranges. Thus, a precise proton range prediction or, at best, an in-situ range control [26] is more required than in the homogeneous irradiation modes like SOBP and HOM. Several techniques for an in-situ range verification that are under current scientific investigation are prompt gamma detection [27], prompt gamma spectroscopy [28], and ionoacoustic range evaluation [29,30].

### 4.2. Proton Minibeam Irradiation Modes

Several experiments in mouse ears have proven that proton minibeam irradiation shows fewer side effects in healthy skin tissue compared to proton broadbeam irradiation of the same mean dose [17,31,32]. In addition, higher survival rates of rats are reported when irradiating their brains by proton (or even carbon ion) minibeams compared to broadbeam irradiation [23,33]. Detailed analysis on neurologic toxicity shows also the tissue sparing capabilities of minibeams [34], although other experiments utilizing heavy ion minibeam irradiation do not show improvements [35]. The reduction of side effects depends on the minibeam sizes [16,17,22] and on the irradiation geometries when fractionating additionally in time [19]. The tissue sparing capabilities of minibeam irradiations are based on various mechanisms of spatial fractionation that reduce the side effects depending on the sort of tissue [33]. While so-called parallel or partially parallel organs like skin may show a strong reduction in side effects, the reduction in side effects by minibeam irradiation of serial organs like brain tissue is still under debate. Nevertheless, there might be a significant potential to reduce the side effects in the healthy tissue and/or enhance tumor control probabilities by proton minibeam or light ion minibeam therapy.

To produce proton minibeams, proton beams are collimated or, in order to obtain better peak-to-valley dose ratios, focused to pencil or planar sub-millimeter-sized beams that are placed at a certain center-to-center (ctc) distance chosen such that a tumor is covered by at least a prescribed tumor dose at any location within the tumor [23,31,33]. A homogeneous tumor dose can be achieved when choosing the right ctc distances for one-sided, two- or more sided irradiation modes [10]. However, tissue sparing through spatial fractionation reduces strongly when close to the tumor.

Critical organs adjacent to the tumor are nowadays the main concern when applying radiation therapy. Sammer et al. [11] have shown that the mean cell survival in the healthy tissue can be enhanced close to the tumor edges when applying interlaced (sometimes also called interleaved) minibeams from two or more sides at a heterogeneous tumor dose. The minimum prescribed tumor dose is obtained at any position in the tumor, but the lateral dose distribution oscillates with a periodicity of the ctc distances. In these studies, the longitudinal dose distribution was based on the spread-out Bragg peak (SOBP) applications where the longitudinal dose distributions from each side were planned to be homogeneous, as it is usually done in conventional proton therapy.

Here, we further follow the route of heterogeneous dose deposition from proton minibeam therapy allowing for an additional longitudinally heterogeneous dose deposition, but maintaining a minimum prescribed tumor dose as previously suggested in [11]. We present distal-edge-enhanced dose deliveries in conjunction with interlaced two-sided proton minibeams set under 180° (Figure 2 and Figure 3) comparing with broadbeam irradiations under the same principle irradiation geometries.

All minibeam irradiation cases investigated in this study yield a much larger mean cell survival than the broadbeam irradiation cases (Figure 4) and thus also show strongly reduced EUD values in the healthy tissue, in particular, close to the tumor edges (Table 1). The best values are obtained for the OPT minibeam mode followed by similar, but slightly lower values for the 1E and HOM minibeam modes. All of these proton minibeam irradiation modes profit from a mean dose reduction in the healthy tissue of the longitudinally heterogeneous dose distributions as in the broadbeam cases, but also from the spatial fractionation effects resulting in dose reduction factors DRF < 0.4 (Table 2) relative to the standard SOBP case. Simultaneously, mean dose levels within the tumor exceed the prescribed tumor dose levels by the substantially enhanced f_D_ ≥ 1.5 (Figure 2 and Figure 3 and Table 2). Both effects imply better tumor control at lower toxicity in the healthy tissue. The OPT minibeam was optimized such that the heterogeneities from spatial fractionation (mainly at the tumor edge) are balanced out by the heterogeneities of the longitudinal dose profile. That is why the lateral minimum dose levels are close to the prescribed tumor dose D_t_ over the entire tumor length. In all other minibeam irradiation modes (SOBP, HOM, and 1E), the minimum value only appears either close to the tumor edges or in the tumor center, resulting in a trade-off for the ctc distances.

In the case of the minibeams, the extreme scenario is the 1E mode where the energy of 146 MeV is chosen such that the distal dose fall-off with D = D_t_ is just at the distal edge of the tumor (see Figure 1). In terms of enhancing mean cell survival, the 1E minibeam mode is only slightly inferior to the optimum minibeam mode (OPT). The dose enhancement in the tumor is large (f_D_ = 1.67), thus increasing the tumor control probability. In addition, the technical realization of the 1E minibeam irradiation would be the easiest when utilizing beam focusing, since every minibeam spot has to be met only once by a single energy. Multiple irradiations of the same location by different energies with submillimeter accuracy are challenging for the other proton minibeam irradiation modes (SOBP, HOM, and OPT).

When considering the spatial fractionation effects from minibeam irradiation, beam size of the minibeams plays a crucial role in their potential of sparing healthy tissue [16,17,22]. At sizes of pencil beams smaller than 2–3 mm nearly no inflammation reaction could be followed in mouse ears, with a gradual increase for larger beam sizes [16]. As crucial size parameters, effective beam diameters d_eff,10%_ and d_eff,50%_ are introduced such that outside this diameter cell survival is larger than 10 % or 50 %, respectively. These effective beam diameters are considered to make inter-comparison of spatial fractionation effects of different application modes possible. They should be as small as possible and below 2 mm at best [16]. Effective beam diameters smaller 2 mm are possible at shallow depths by beam collimation or focusing. However, they widen due to lateral spread at larger depth. Since the lateral spread of proton minibeams is reduced at higher proton energies the 1E, HOM, and OPT minibeam irradiation modes show lower effective beam sizes than the SOBP minibeam cases close to the tumor edges (Table 2). The effective beam sizes are slightly larger than those at which zero reactions can be expected in the healthy tissue, but still in a range where reduced side effects can be expected. Since the minibeams spread with depth, much smaller diameters and thus even better tissue sparing will occur at smaller depths. The differences in tissue sparing between planar or pencil beams are also not well established. The beam size and form effects will have to be studied in the future for different tissues.

Ion beams like helium or carbon ion beams provide smaller lateral spread than the proton beams considered here [22,36,37]. Similar irradiation modes as HOM, 1E, and OPT may be introduced for these heavy ion beams by scaling the ctc distances. However, some differences will show up in detail since the heavier ion beams exhibit sharper Bragg peaks, and some dose is obtained behind the Bragg peak due to spallation. A best compromise of smaller minibeam sizes close to the tumor and with a low dose behind the tumor may be helium beams.

### 4.3. 1E and Proton Minibeam Irradiation Modes in Conjunction with FLASH

Several preclinical experiments have demonstrated that tissue toxicity can be reduced by the so-called FLASH irradiation modes, but maintain similar tumor control when the dose is deposited at high dose rates (>10 Gy/s) [38,39,40,41]. Reduced toxicity is reported mainly at larger single fraction dose values above ca. 8 Gy [42]. In any kind of current proton therapy centers, the combination of FLASH conditions with the preferred pencil beam scanning method and common SOBP irradiation mode is hardly realizable. In pencil beam scanning, the lateral irradiation field is formed by an overlap from several adjacent pencil beams applied one after the other. It has been shown by analyzing optimized irradiation plans that the maximum portion of a single dose spot is 20% or even less at any place in the tumor and many beam positions and energies are required to meet the full dose [43]. Thus, the delivered dose is always split into several beams applied with some intrinsic time separation limiting the mean dose rate at any position in the healthy tissue where the FLASH effect is intended to be utilized. Therefore, an ultra-fast beam scanning and modulation in energy is necessary for FLASH irradiations in conjunction with pencil beam scanning, which is technically very challenging.

An alternative for ultra-fast beam delivery for utilizing FLASH effects may be obtained by the 1E proton broadbeam irradiation mode. Here, only one energy is applied when irradiating from one side. Thus, the challenge is reduced to deliver the dose by fast beam scanning of this single beam energy in less than about a second from one direction. This will require high beam currents of many up to several hundred nanoamperes and a fast scanning unit, but may be technically feasible.

An alternative for FLASH application comes with the interlaced minibeam irradiations. The minibeams, as considered in the study, are laterally well separated in the healthy tissue up to the tumor edges. As a consequence, each position of high dose in the healthy tissue is met by only one minibeam. The energy modulation can be decreased by using only a few (HOM and OPT), or in extreme cases only one proton energy (1E mode). If energy modulation is necessary, fast beam energy change may be offered by new accelerator technologies [44,45]. The 1E irradiation mode may be possible even with conventional proton accelerators that are able to deliver small planar minibeams [46]. In addition, a fast dose delivery as aimed for in the FLASH proton minibeam therapy will keep a spread-out of the minibeams from tissue movements negligible. Thus, it might be a prerequisite for successful proton minibeam application to combine proton minibeams with FLASH applications as suggested by Reindl and Girst [47].

In addition, the peak dose delivered by a minibeam into healthy tissue is usually much higher than the prescribed dose in the tumor. Thus, the threshold values for FLASH being in the range of 8 Gy are easily applied even if the mean dose in the tumor is kept lower, combining fractionation in time, spatial fractionation by minibeams, longitudinally heterogeneous dose delivery, and FLASH irradiation.

## 5. Conclusions

Longitudinally heterogeneous dose distributions are investigated in terms of their potential to reduce the side effects in proton broadbeam and proton mininbeam irradiation modes. The main assumption is that the upper dose limit within the tumor can be disregarded as long as the side effects are reduced by a more heterogeneous dose delivery in the tumor while maintaining the minimum prescribed tumor dose at any position in the tumor. The calculations show that substantially lower dose deposition in the healthy tissue is obtained and thus a higher cell survival is expected, when using longitudinally heterogeneous dose distributions as in the case of proton broadbeam irradiation modes. The advantage of proton minibeam over all proton broadbeam irradiation modes is documented by an even further enhanced mean cell survival in the healthy tissue, in particular, directly in front of the tumor.

Overlaid longitudinally heterogeneous proton irradiation fields may be applied in similar ways as described here for any kind of deep-seated tumor that can be tackled by proton beams from two opposing sides. Extended irradiation plans utilizing irradiation fields from two non-opposing sides, or even more than two irradiation fields, are possible, but have to be adapted to any real tumor location. Tumor sites, where the longitudinally heterogeneous irradiation modes bear the potential of reduced toxicity in the healthy tissue and enhanced tumor control probabilities, may be for example, tumors in the brain, liver, pancreas, prostate, and lung. All these entities may profit from the combination of longitudinally and laterally inhomogeneous dose distributions as suggested here for interlaced proton minibeams.

Since the spatial fractionation effect of particle minibeams depends also on the minibeam effective sizes, heavier ion beams like helium or carbon ion minibeams may be favored. More experimental data on tissue sparing capabilities are needed to fully exploit tissue sparing by proton or heavier ion minibeam therapy. These new longitudinally heterogeneous minibeam therapy modes may also pave the way to reduce the technical efforts in minibeam irradiation scenarios. In addition, they may offer the combination of minibeam and FLASH to achieve the highest possible sparing of healthy tissues.

## Figures and Tables

**Figure 1 cancers-14-05162-f001:**
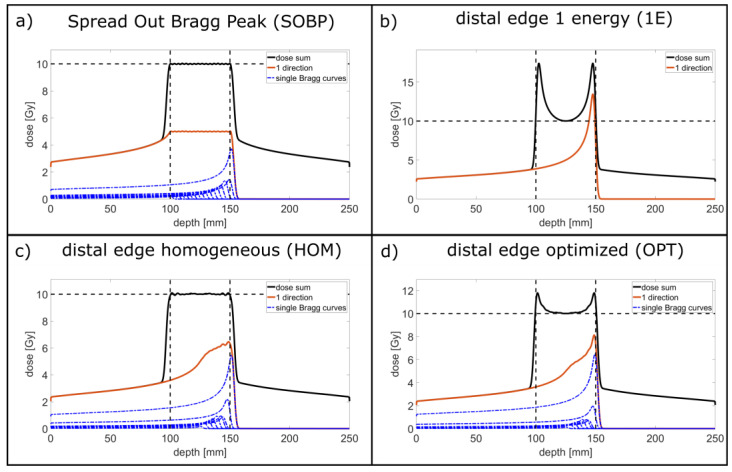
Depth-dose curves for the four considered longitudinal irradiation modes. (**a**) Spread Out Bragg Peak (SOBP). (**b**) distal edge 1 energy (1E). (**c**) distal edge homogeneous (HOM). (**d**) distal edge optimized (OPT). The blue dashed–dotted lines mark the single Bragg curves being the depth-dose distributions from one-sided irradiations of one proton energy that are optimized in height to achieve the summed depth-dose curves plotted as black lines. The red curves describe the sum of all single Bragg curves from one direction. The dashed lines mark the tumor edges (vertical) and the prescribed dose of 10 Gy (horizontal).

**Figure 2 cancers-14-05162-f002:**
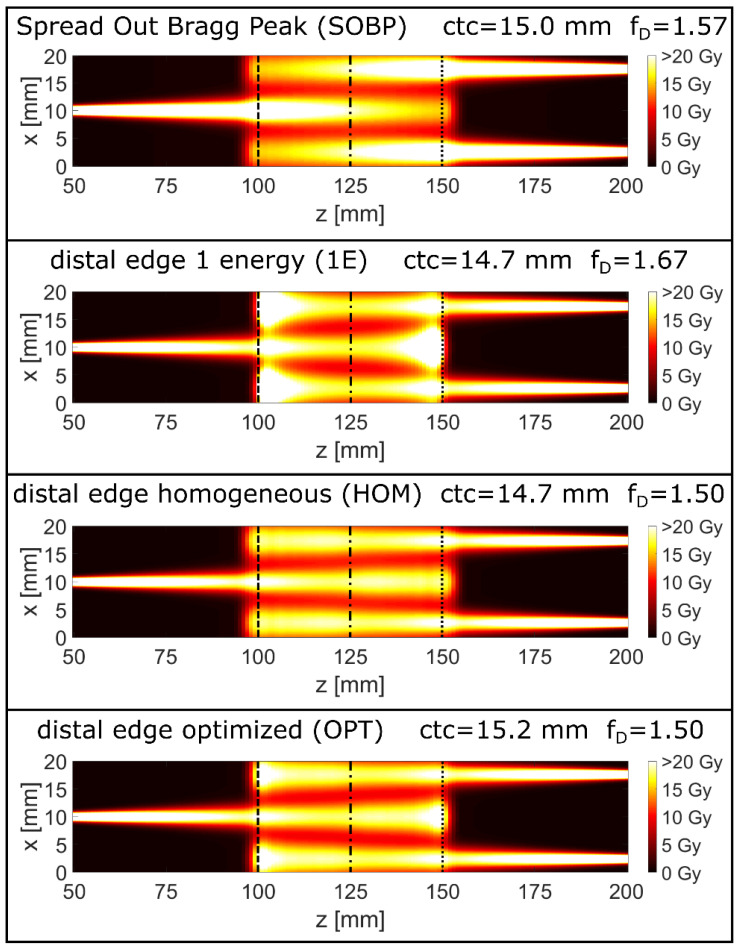
Dose maps for a longitudinal range of 50–200 mm and a lateral range of 20 mm of 2-directional, interlaced planar proton minibeams optimized for the different depth-dose distributions. The dose is color-coded with a saturation at 20 Gy. The dashed and the dotted lines mark the tumor edges at 100 mm (dashed) and 150 mm (dotted). The dashed–dotted lines mark the tumor center at 125 mm.

**Figure 3 cancers-14-05162-f003:**
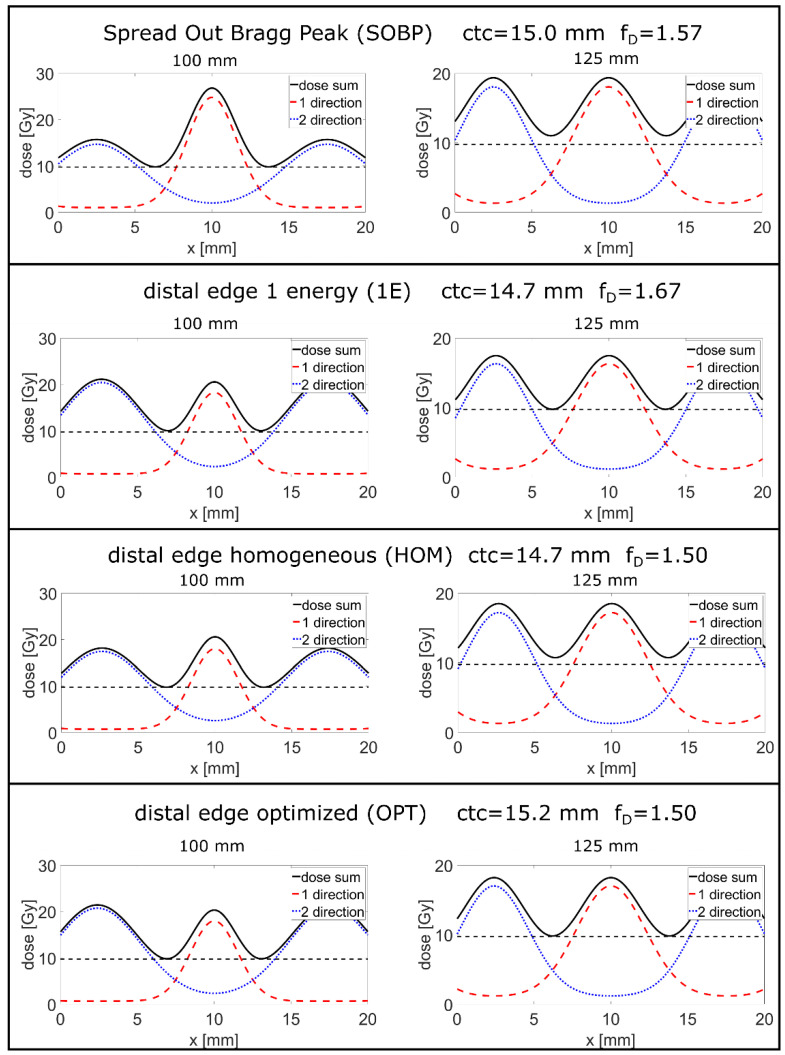
Cross-sections of the dose distributions of the different interlaced minibeam irradiation modes at the tumor edge at a depth 100 mm (the dashed line in Figure 2) and the tumor center at 125 mm (dashed–dotted lines in Figure 2). The dashed red lines are the cross sections of dose distributions for the irradiation applied from one direction (from left to right in Figure 2) and the dotted blue lines are coming from the opposite direction (from right to left in Figure 2). The solid black line is the dose sum of direction 1 and direction 2. The dashed black lines mark the minimum dose criterion of 0.975 × D_t_ = 9.75 Gy.

**Figure 4 cancers-14-05162-f004:**
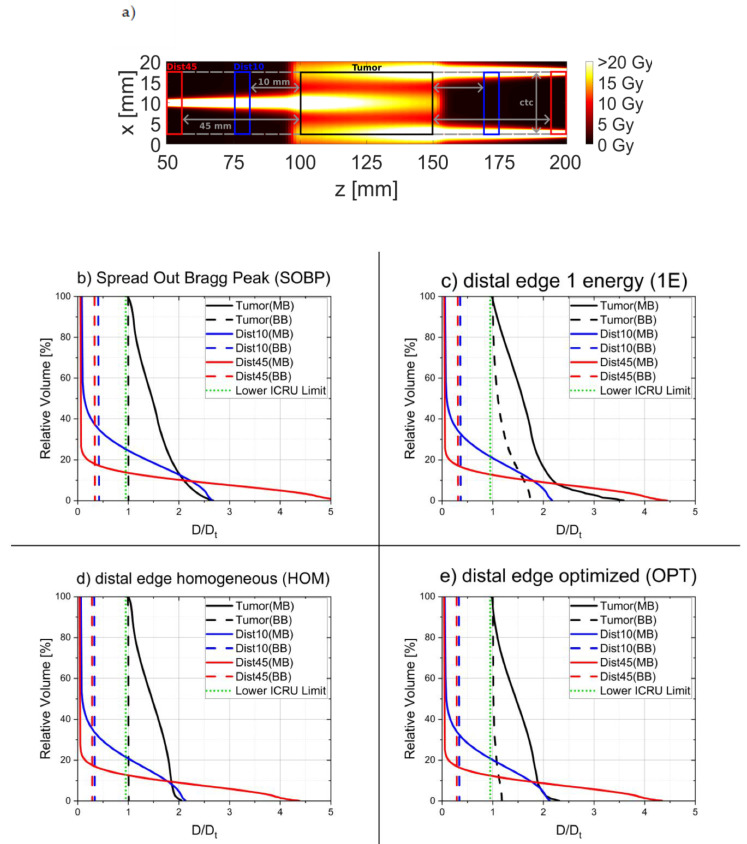
Dose-volume histograms (**b**–**e**) of the regions marked in (**a**) for the four considered longitudinal irradiation modes, for both the broadbeam (BB) and minibeam (MB). The lower dose limit of D/D_t_ = 0.975in the tumor is displayed in green.

**Figure 5 cancers-14-05162-f005:**
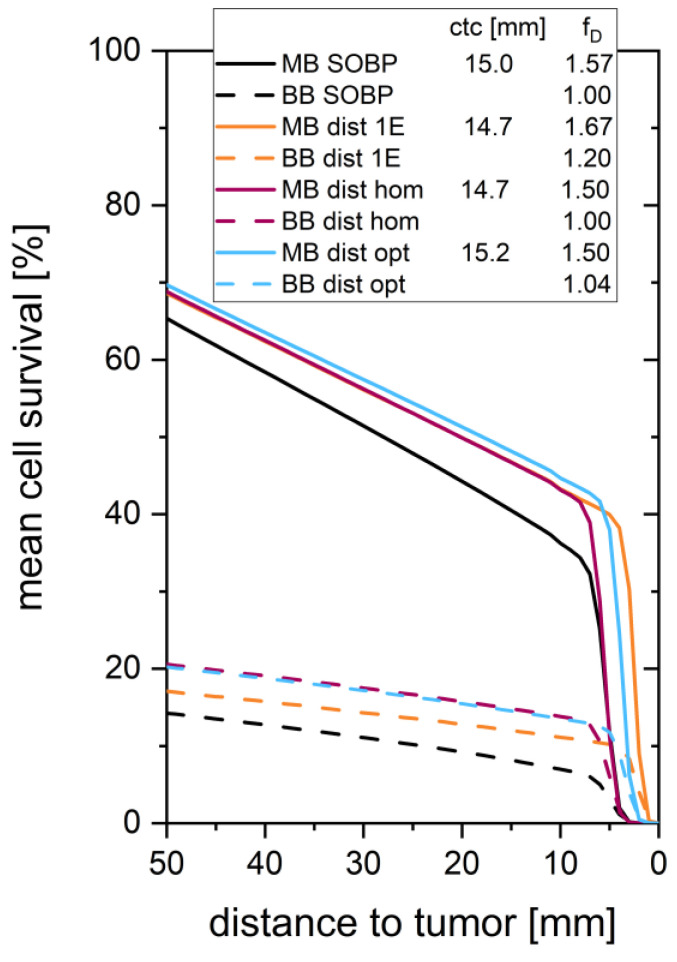
Mean cell survival over the distance to tumor for a prescribed tumor dose D_t_ = 10 Gy (solid lines for interlaced minibeam irradiation (MB), dashed lines for broadbeam irradiation (BB)). The different applied depth-doses are color marked with black, orange, purple, and blue for SOBP, 1E, HOM, and OPT, respectively.

**Table 1 cancers-14-05162-t001:** Proton energies simulated for 4 different longitudinal irradiation modes. The table lists the used energies and the corresponding relative weighting factors for the depth-dose curves from one direction. Note that the relative weighting factors are only valid within the same depth-dose curve and cannot directly be compared between the scenarios.

**SOBP**	Energy [MeV]	117	119	121	123	125	127	129	131	133
	Weighting factor	0.077	0.081	0.085	0.093	0.101	0.105	0.113	0.125	0.137
	Energy [MeV]	135	137	139	141	143	145	147	148	
	Weighting factor	0.149	0.169	0.190	0.226	0.266	0.363	0.387	1	
**1E**	Energy [MeV]	146								
	Weighting factor	1								
**HOM**	Energy [MeV]	133	135	137	139	141	143	144	146	148
	Weighting factor	0.026	0.066	0.105	0.128	0.162	0.167	0.155	0.397	1
**OPT**	Energy [MeV]	133	135	137	139	141	143	144	146	147
	Weighting factor	0.018	0.051	0.077	0.090	0.118	0.118	0.109	0.300	1

**Table 2 cancers-14-05162-t002:** Characteristics of the different irradiation modes.

Irradiation Mode	ctc [mm]	f_D_	Depth = 90 mm and 160 mm, thus, 10 mm from Tumor Edges D_t_ = 10 Gy						
			D_mean_ [Gy]	S [%]	EUD [Gy]	DRF_SOBP_	d_eff,10%_ [mm]	d_eff,50%_ [mm]	Δ [mm]
Broad SOBP	-	1.00	4.23	7.0	4.23	1	-	-	4.6
Broad 1E	-	1.20	3.66	11.1	3.66	0.87	-	-	2.1
Broad HOM	-	1.00	3.37	13.8	3.37	0.80	-	-	5.0
Broad OPT	-	1.04	3.40	13.5	3.40	0.80	-	-	3.2
MB SOBP	15.0	1.57	4.23	36.2	1.96	0.46	5.7	7.7	4.6
MB 1E	14.7	1.67	3.66	43.2	1.66	0.39	4.9	6.6	2.1
MB HOM	14.7	1.50	3.37	43.1	1.67	0.39	5.0	6.6	5.0
MB OPT	15.2	1.50	3.40	44.7	1.60	0.38	5.0	6.6	3.2

## Data Availability

The data presented in this study are available on request from the corresponding author.

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
