# Peer review of "Longitudinally Heterogeneous Tumor Dose Optimizes Proton Broadbeam, Interlaced Minibeam, and FLASH Therapy"

_cancers, 2022, doi:10.3390/cancers14205162_

Round 1

Reviewer 1 Report

In their article 'Longitudinally heterogeneous tumor dose optimizes proton broadbeam, interlaced minibeam and FLASH therapy' Sammer and colleagues present their work to reduce the dose exposure of the healthy tissue during proton beam therapy by extended heterogeneous dose deposition while maintaining the prescribed minimum dose level in the tumor volume. Due to the steadily increasing number of long-term cancer survivors who are at increased risk for late sequelae of radiotherapy, further reduction of dose exposure to normal tissue is still of high relevance.

This is a very well-conducted study with appropriately good methodology on this topic. The results are clearly presented and the conclusions are comprehensible. There are no further comments or suggestions for improvement from my side and I recommend the publication in its present form.

Author Response

Thank you very much for appreciating our work as a whole.

Reviewer 2 Report

The paper deals with longitudinally heterogeneous proton irradiation modes. The results show a dose reduction in the healthy tissue compared to the common SOBP mode in case of broad proton beams.

The authors concluded: "The calculations show that substantially higher cell survival is obtained in the healthy tissue already when using longitudinally heterogeneous dose distributions in case of proton broadbeam irradiation modes." However experimental support is recommended in order to verify this statement.

Author Response

Thank you for this comment. According to the lower dose deposition in the healthy tissue as simulated in case of irradiation modes 2 to 4 a higher cell survival can be expected due to the well-established survival-dose relationships that are based on experimental data. At the present stage of research, we think that an experimental approval is not necessary since sufficient experimental data are available for this general statement. In order to clear up the statement we have modified the sentence

to: “The calculations show that substantially lower dose deposition in the healthy tissue is obtained and thus a higher cell survival is expected already when using longitudinally heterogeneous dose distributions in case of proton broadbeam irradiation modes.”

Reviewer 3 Report

Dear authors, thank you for your effort and endeavor. This is a study on innovative issues that may intrigue any one working in the cancer field. FLASH radiotherapy is defined as single ultra-high dose-rate (≥ 40 Gy/s) radiotherapy. Many would like to learn the latest about proton and FLASH and how different modes should inform treatment selection and impact patient outcomes. When Radiation Oncologists work with their multidisciplinary teams to guide clinical decisions, they perform radiotherapy to ensure accurate dose delivery on best practices. I am appreciative of such privilege to review your article.

1.       Overall, do not use abbreviation without first indicating the whole term.

2.       In Line 47, the reference for “the general assumption” was lacking.

3.       In Line 60, please be more specific about “much higher” mean cell survival…. How much higher?

Author Response

Thank you very much for your appreciation. We tried to improve the English language and style

 Overall, do not use abbreviation without first indicating the whole term.

We have added the indications to abbreviations where they were missing 

  1. In Line 47, the reference for “the general assumption” was lacking.

We have added a reference where this consideration has been quantitatively described:

  1. Brahme, Dosimetric precision requirements and quantities for characterizing the response of tumors and normal tissues, in IAEA-TECDOC-896, Radiation dose in radiotherapy from prescription to delivery, 1996, ISSN 1011-4289, IAEA, Austria 1996
  2. In Line 60, please be more specific about “much higher” mean cell survival…. How much higher?

We have changed the sentence to

“When combined with additionally lateral dose heterogeneities in proton minibeam therapy, up to 7 times larger mean cell survival and a corresponding reduction by a factor smaller than 0.4 of the applied effective unified dose is obtained in the healthy tissue compared to all broadbeam cases. The investigated longitudinally inhomogeneous dose distributions even enhance the potential to reduce side effects of proton minibeam irradiation modes.

Reviewer 4 Report

Congratulations to the authors for this manuscript, which represents a clear advance in cancer treatment therapies.

The manuscript is well written, has very well elaborated graphs and figures, is orderly and very well structured.

The only thing missing is at least mention of other treatments to combat cancer, such as immunotherapies, ... and indicate the types of cancer in which the treatments described by the authors are more effective. In this way, the current situation regarding types of cancer in which it is more effective to apply the techniques described by the authors would be better placed for potential readers, so it would need a minor revision.

Author Response

The manuscript is well written, has very well elaborated graphs and figures, is orderly and very well structured.

Thank you very much for your very positive judgement

The only thing missing is at least mention of other treatments to combat cancer, such as immunotherapies, ... and indicate the types of cancer in which the treatments described by the authors are more effective. In this way, the current situation regarding types of cancer in which it is more effective to apply the techniques described by the authors would be better placed for potential readers, so it would need a minor revision.

We have included one sentence at the beginning of the introduction to set radiotherapy as such into perspective:

Beside surgery, chemotherapy and immunotherapy, radio therapy is one of the main treatment options of solid tumors being applied in about 50 % of all cases [Borras16, Delaney15].

In addition, we have included the following paragraph into the conclusion section in order to better define the potential of the described therapy option:

Overlaid longitudinally heterogeneous proton irradiation fields may be applied in similar ways as described here for any kind of deep-seated tumor that can be tackled by proton beams from two opposing sides. Extended irradiation plans utilizing irradiation fields from two non-opposing sides or even more than two irradiation fields are possible but have to be adapted to any real tumor location. Tumor sites, where the longitudinally heterogeneous irradiation modes bear the potential of reduced toxicity in the healthy tissue and enhanced tumor control probabilities may be tumors in the brain, liver, pancreas, prostate and lung. All these entities may profit in particular from the combination of longitudinally and laterally inhomogeneous dose distributions as suggested here for interlaced proton minibeams.

Round 2

Reviewer 3 Report

Congratulations on your hard work!

And thank you for making your work clearer

for the readers to comprehend!